# A Novel LiDAR–IMU–Odometer Coupling Framework for Two-Wheeled Inverted Pendulum (TWIP) Robot Localization and Mapping with Nonholonomic Constraint Factors

**DOI:** 10.3390/s22134778

**Published:** 2022-06-24

**Authors:** Yanwu Zhai, Songyuan Zhang

**Affiliations:** State Key Laboratory of Robotics and System, Harbin Institute of Technology, Harbin 150001, China; zslhit@126.com

**Keywords:** Lidar-IMU-Odometer system, two-wheeled inverted pendulum robot, ground constraints, nonholonomic constraint factor

## Abstract

This paper proposes a method to solve the problem of localization and mapping of a two-wheeled inverted pendulum (TWIP) robot on approximately flat ground using a Lidar–IMU–Odometer system. When TWIP is in motion, it is constrained by the ground and suffers from motion disturbances caused by rough terrain or motion shaking. Combining the motion characteristics of TWIP, this paper proposes a framework for localization consisting of a Lidar-IMU-Odometer system. This system formulates a factor graph with five types of factors, thereby coupling relative and absolute measurements from different sensors (including ground constraints) into the system. Moreover, we analyze the constraint dimension of each factor according to the motion characteristics of TWIP and propose a new nonholonomic constraint factor for the odometry pre-integration constraint and ground constraint factor in order to add them naturally to the factor graph with the robot state node on SE(3). Meanwhile, we calculate the uncertainty of each constraint. Utilizing such a nonholonomic constraint factor, a complete lidar–IMU–odometry-based motion estimation system for TWIP is developed via smoothing and mapping. Indoor and outdoor experiments show that our method has better accuracy for two-wheeled inverted pendulum robots.

## 1. Introduction

In recent years, Simultaneous localization and mapping (SLAM) for intelligent mobile robots has become a research hotspot [1,2]. Under GPS-denied and unknown environment conditions, mobile robots must be able to utilize onboard sensors to construct an environment map that can be used for navigation [3] and then rely on this environment map for navigation, collision avoidance, and path planning. Many solutions have been proposed based on different sensing methods, mainly divided into vision-based [4,5,6], lidar-based [7,8,9], and a combination of the two [10,11]. Meanwhile, in order to cope with complex environments, sensors such as IMUs and encoders that are less affected by the environment can be used to assist with visual [12,13] or lidar [14,15,16,17,18] localization. Although vision-based methods are particularly suitable for position recognition, their sensitivity to initialization, illumination, and range as well as their high depth computation cost makes them unreliable when supporting autonomous navigation systems.

On the other hand, lidar can directly obtain depth information about the surrounding environment, and lidar are immune to illumination changes. Many impressive LiDAR-based localization and mapping algorithms have been proposed. In these algorithms, the IMU, encoder and other high-frequency output sensors are usually used to assist with lidar localization and the deskew point cloud. In [7], the authors present lidar odometry and mapping (LOAM) for low-drift real time state estimation and mapping. In this algorithm, the IMU is only used to de-skew the point cloud and does not participate in lidar localization. Here, for the localization algorithm of the lidar–IMU system, we divided it into filter-based and optimization-based segments according to the coupling method. Filter-based methods typically utilize extended Kalman filtering to couple measurements from lidar and IMU to estimate the state of the robot frame-by-frame. In [14], Lynen et al. proposed a modular method that uses EKF to fuse IMU measurements with relative pose measurements from cameras, lidars, etc. In [15], the authors present a lidar-inertial state estimator called R-LINS, which uses a recursive error-state Kalman filter to estimate the robot’s state. In [16], the authors propose a fast, robust and general LiDAR–IMU odometry framework based on an efficient and tightly-coupled iterative Kalman filter for fast, robust, and accurate LiDAR navigation. Optimization-based methods usually use nonlinear optimization to combine a fixed number of lidar keyframes and IMU pre-integration in order to estimate the optimal pose. In the literature [17], a tightly coupled framework lio-mapping of lidar and IMU has been introduced which uses a framework similar to VINS and achieves good accuracy; however, it cannot run in real time. In [18], the authors proposed a framework LIO-SAM for tightly coupling lidar and IMU via smoothing and mapping, achieving highly accurate real-time localization and mapping. Filtering-based methods usually only consider the influence of the previous frame on the current frame, which incurs lower computational costs. The optimization-based method usually adopts the sliding window method to evaluate the constraint relationship between the robot states of the previous n frames (including the current frame). The optimization-based method is more computationally expensive than the filtering-based method, however, it has accuracy. With improved hardware performance, optimization-based methods can run in real time, and should gradually become the mainstream of research. However, in scenarios with sparse structural features, such as long tunnels, wide roads, etc., lidar lacks constraints in certain directions, resulting in decreased localization accuracy. In [19], the authors found that when their mobile robot had constant acceleration or no rotation, the VINS system was unable to distinguish the magnitude of the true body acceleration and the direction of the local gravitational acceleration from that of the accelerometer bias. Similar situations can appear in scenarios where Lidar is degraded.

The localization of mobile ground platforms with unique system architectures has not previously been thoroughly investigated and discussed. Due to ground constraints, many vision-based [20,21] and lidar-based [22,23] SLAM algorithms use the SE(2) pose. However, the ground is rough in the natural environment, and a ground-mobile robot will be disturbed by motion disturbance. In order to better adapt to the natural environment, Ref. [24] proposes a new constraint model, stochastic SE(2) constraints, which usually parameterizes the robot pose on SE(3), allowing for slight disturbance in dimensions other than SE(2). Fan Zheng [25] extended the work of the above article and proposed a complete motion estimation system that utilizes SE(2) constraints and SE(3) pose parameterization.

In this paper, a factor graph is formulated to couple the lidar, IMU, and encoder measurements for the localization and mapping of TWIP robots on an approximately flat road surface. Inspired by the SE(2) constraint–SE(3) pose method used in [24], we propose a new factor with constraints to introduce the encoder’s pre-integration and ground constraints, which we achieve by analyzing the motion profile of the robot and its deviation from planar motion (e.g., due to terrain unevenness or motion vibration) and formulating stochastic (i.e., “soft”) instead of deterministic (i.e., “hard”) constraints. This allows us to properly model the TWIP’s almost-planar motion, as shown in Figure 1. Moreover, we use Lie algebra to represent the rotation in the pose imitation process. Lie algebra has no redundant parameters. Compared with Euler angles, Lie algebra does not have gimbal locking problems, and rotation integration based on Lie algebra can be integrated into a closed form, whereas Euler angles are only exact up to the first order [26]. The main novel contributions of this work are as follows:A complete LIDAR–IMU–encoder-coupled state estimation and mapping framework is proposed for a two-wheeled inverted pendulum robot. This method extends the LIDAR–IMU to process odometric measurements and improves the system’s localization accuracy.A new nonholonomic factor is proposed to introduce encoder pre-integration constraints and ground constraints, which are naturally coupled to the factor graph optimization of the system.Indoor and outdoor experiments demonstrate the improved localization accuracy and robustness of the proposed lidar–IMU–odometry coupling framework when the controller is mounted on a two-wheeled inverted pendulum robot navigating on an approximately flat surface.

The rest of the paper is organized as follows: Section 2 provides an overview of the entire methodology; Section 3 reviews the basic knowledge which is be used later in the paper; Section 4 details the five types of factors introduced in our algorithm; and Section 5 describes several experiments we conducted to demonstrate the effectiveness of the proposed method.

## 2. Methodology Overview

This paper aims to estimate the motion state of a two-wheeled inverted pendulum robot (TWIP) via lidar–IMU–Odometry coupling. Compared with a six-degrees-of-freedom robot, TWIP cannot move sideways, and is constrained by the ground. Compared with autonomous vehicles on the ground, TWIP swings back and forth during movement to maintain the balance of movement. If we directly parameterize the robot pose on SE(3) without considering the ground constraints, the state estimation accuracy and robustness is depressed due to fewer constraints in the vertical direction of the lidar. If only ground constraints are considered, that is, the deterministic constraint, the performance of system motion estimation is degraded because the six dimensions of the pose are highly coupled with sensor observations. Therefore, neither accurate pose parameterization considering ground constraints nor 3D pose parameterization can satisfactorily represent the pose of the ground robot, because the former suffers from out-of-constraint motion while the latter does not use plane motion constraints. This prompted us to study the state estimation of the two-wheeled inverted pendulum robot through a lidar–IMU–Odometry system.

This paper proposes a framework for localization using a Lidar–IMU–Odometer system. The flow chart of this framework is shown in Figure 2. When the system receives a lidar scan, it first uses the measurement of the IMU to de-skew the point cloud, then extracts the line and surface feature points. If the pose exceeds a pre-set threshold, the current frame is selected as a keyframe; otherwise, it is discarded. Measurements from the IMU provide the initial pose. Then, the keyframes are matched to the global map to obtain the rough global pose and construct the lidar pose factor. The IMU and odometer measurements between two adjacent lidar keyframes are used to construct constraint factors via pre-integration techniques. Meanwhile, we use the random constraint model to construct the ground constraint factor. Finally, all factors are included in the factor graph and jointly optimized to obtain the optimal robot pose. The factor graph formulated for TWIP, which is shown in Figure 3, incorporates the measurement data of each sensor as a factor in our optimization system. In addition to the above four factors, this factor graph contains closed-loop factors grouped into two categories. The IMU pre-integration factor, the lidar pose factor, and the loop closure factor constrain each dimension of the robot pose, and are named holonomic constraint factors. However, the odometry pre-integration factor and the ground constraint factor only constrain part of the dimensions of the robot’s pose, and thus are named non-holonomic constraint factors. For these factors we propose a nonholonomic constraint factor to add them to the system, inspired by the SE(2) constraint SE(3) parameterization method used in [24]. Utilizing such a constraining factor, a complete lidar–IMU–odometry-based motion estimation system for TWIP is developed via smoothing and mapping.

## 3. Preliminaries

This paper formulates the state estimation problem based on the Lidar–IMU–Odometry system in terms of a factor graph, that is, a nonlinear optimization problem involving the quantities existing on smooth manifolds. Before delving into details, we review several of the related practical geometric concepts.

### 3.1. On-Manifold Pose Parameterization

We usually use the Matrix Lie group to describe three-dimensional rotation, which is defined as follows:(1)$$\mathrm{SO}\left(3\right)=\left\{R\in R^{3x3},R^{T}R=I\right\}$$

However, the rotation combination operation in the SO(3) involves matrix multiplication in the optimisation process, which cannot be directly operated as Euclidean vectors can. Due to Lie groups having the properties of differential manifolds, the tangent space to the manifold (at the identity), called the Lie algebra, can be introduced as a representation of the most diminutive vector form, and is related to Lie groups through matrix exponents and logarithms. Lie algebra is defined as the follows:(2)$$so\left(3\right)=\left\{\phi \in R^{3},\Phi =\phi^{\wedge }\in R^{3\times 3}\right\}$$
where ϕ is the rotation vector of $R^{3}$, corresponding to a 3 × 3 skew-symmetric matrix. The hat operator ${\left(•\right)}^{\wedge }$ is used to find the antisymmetric matrix of a vector, and its form is
(3)$$\Phi =\phi^{\wedge }=\left[\begin{array}{ccc} 0 & -\phi_{3} & \phi_{2}\\ \phi_{3} & 0 & -\phi_{1}\\ -\phi_{2} & \phi_{1} & 0 \end{array}\right]\phi =\left[\begin{array}{c} \phi_{1}\\ \phi_{2}\\ \phi_{3} \end{array}\right]$$

At the same time, we define the inverse mapping of the hat operator ${\left(•\right)}^{\vee }$ as follows:(4)$$\phi =\left[\begin{array}{c} \phi_{1}\\ \phi_{2}\\ \phi_{3} \end{array}\right]=\Phi^{\vee }$$

For any two 3D vectors r and s and the rotation matrix R, the skew-symmetric matrix has the following properties:(5)$$r^{\wedge }s=-s^{\wedge }r$$
(6)$${\left(\mathrm{Rs}\right)}^{\wedge }=Rs^{\wedge }R^{T}$$
(7)$${\left(r^{\wedge }\right)}^{T}=-r^{\wedge }$$
(8)$$r^{\wedge }r^{\wedge }=rr^{T}-I_{3\times 3}$$

Lie algebra can be transformed into a Lie group by exponential mapping (Rodriguez formula):(9)$$\exp\left(\phi^{\wedge }\right)=I+\frac{\sin\left(∥\phi ∥\right)}{∥\phi ∥}\phi^{\wedge }+\frac{1-\cos∥\phi ∥}{{∥\phi ∥}^{2}}{\left(\phi^{\wedge }\right)}^{2}$$

A first-order approximation of the exponential map used later on is
(10)$$\exp\left(\phi^{\wedge }\right)\approx I+\phi^{\wedge }$$

In the optimization process, we often use a left-multiplicative perturbation on the manifold through the left Jacobian matrix, approximating a local on-manifold transformation with Lie algebra:(11)$$\exp\left(\delta \phi^{\wedge }\right)\exp\left(\phi^{\wedge }\right)=\exp\left({\left(J_{l}^{-1}\left(\phi \right)\delta \phi +\phi \right)}^{\wedge }\right)$$
where the left Jacobian matrix is
(12)$$J_{l}\left(\phi \right)=\frac{\sin\theta }{\theta }I+\left(1-\frac{\sin\theta }{\theta }\right)aa^{T}+\frac{1-\cos\theta }{\theta }a^{\wedge }$$
where θ and a are the norm and direction of the rotation vector ϕ, ϕ=θa. The inverse of the left Jacobian matrix is
(13)$$J_{l}^{-1}\left(\phi \right)=\frac{\theta }{2}\cot\frac{\theta }{2}I+\left(1-\frac{\theta }{2}\cot\frac{\theta }{2}\right)aa^{T}-\frac{\theta }{2}a^{\wedge }$$

Another valuable property of the exponential map is
(14)$$R\mathrm{Exp}\left(\phi \right)R^{T}=\exp\left(R\phi^{\wedge }R^{T}\right)=\mathrm{Exp}\left(R\phi \right)$$

### 3.2. Factor Graph Optimization

As TWIP moves, lidar measurements are continuously collected to estimate the robot’s motion state and model the surrounding environment. This process can be described by a factor graph, where the robot states to be estimated are state nodes. The continuously collected measurements are introduced as factor nodes connected by edges to the involved state nodes (see Figure 3). In general, we assume that these measurements are independent of each other and are affected by Gaussian noise. The robot state estimation problem represented by the factor graph above can be formulated for maximum a posteriori estimation. Therefore, we formulate the above problem as the following nonlinear least-squares problem:(15)$$\hat{x}=\arg\min_{x}\sum_{i}{∥r_{i}\left(x\right)∥}_{\sum_{i}}^{2}$$
where $r_{i}$ is the zero-mean residual associated with measurement *i*, x is the parameter vector to be optimized, $\Sigma_{i}$ is the measurement covariance, and ${∥r_{i}∥}_{\Sigma_{i}}^{2}=r_{i}^{T}\Sigma_{i}^{-1}r_{i}$ is the energy norm.

The above least squares problem can be solved iteratively using the Gauss–Newton method. At each GN iteration, we approximate the optimized objective function (the residual function) by first-order Taylor expansion at the current estimated state ${\hat{x}}^{-}$:(16)$$\begin{array}{c} \Delta \hat{x}=\arg\min_{\Delta x}\sum_{i}{∥r_{i}\left(x⊕\Delta x\right)∥}_{\sum_{i}}^{2}\\ \;\;\;\;\;=\;\arg\min_{\Delta x}\sum_{i}{∥r_{i}\left(x\right)+H_{i}\Delta x∥}_{\sum_{i}}^{2} \end{array}$$
where $H_{i}=\frac{\partial r_{i}}{\partial x}$ is the Jacobian matrix of the ith residual for the robot state and Δx is the increment of the state obtained at each iteration. We define the generalized update operation, ⊕, which maps a change in the error state to one in the entire state. After solving the linearized equation, the current state is updated as ${\hat{x}}^{+}={\hat{x}}^{-}⊕\Delta x$. This linearization process is then repeated until convergence

## 4. State Estimation with Accurate Parameterization of TWIP

### 4.1. Pose Parameterization of TWIP

Compared with a six-degrees-of-freedom robot, TWIP cannot move sideways and is constrained by the ground. Compared with autonomous vehicles on the ground, TWIP swings back and forth during movement to maintain the balance of movement. Therefore, we cannot characterize its motion state completely in SE(2) or SE(3) space. A simplified diagram of this movement is shown in Figure 4.

In order to analyze the movement of TWIP more conveniently, its coordinate system is defined as follows:

Lidar coordinate system (L): The lidar frame originates at the lidar center. The *y*-axis points to the left, the *z*-axis points upward, and the *x*-axis points forward, coinciding with the positive direction of TWIP.

IMU coordinate system (I): The IMU is located below the lidar; its frame originates at the IMU measurement center. The *x*-, *y*-, and *z*-axes are parallel to (L), pointing in the exact directions, and the *z*-axis points forward, coinciding with the *z*-axis of the lidar frame.

Robot body coordinate system (B): As shown in Figure 4, the robot body frame is connected to the robot body during the movement process. Its origin is at the midpoint of the line connecting the centers of the two wheels. The *x*-, *y*-, and *z*-axes are parallel to (L), pointing in the exact directions, and the *z*-axis points forward, coinciding with the *z*-axis of the lidar frame.

World coordinate system (W): The world frame is the gravity-aligned coordinate system, which is initially coincident with the robot body coordinate system (B).

All of the above frames follow the right-hand rule. We calibrate the transformation between the lidar frame and the robot body frame and between the IMU frame and the robot body frame. The transformation between the robot body coordinate system and the world coordinate system is the motion state of the robot, that is, the target parameters for optimization. Figure 5 shows the posture of TWIP as obtained by the Attitude and Heading Reference System (ARHS) during movement. As seen from the figure, when TWIP moves, there are movements on the two-dimensional plane (yaw) and front and back swings (pitch), as well as movement disturbances in the roll direction. Therefore, we parameterize the robot’s pose on SE(3). The motion state of the robot corresponding to the lidar keyframe $F_{i}$ at time *i* is
(17)$$K_{i}=\left[\begin{array}{cc} R_{B_{i}}^{W} & P_{B_{i}}^{W} \end{array}\right]$$
where $P_{B_{i}}^{W}\;=\;{\left[\begin{array}{ccc} x_{i} & y_{i} & z_{i} \end{array}\right]}^{T}$ is the translation and $R_{B_{i}}^{W}\in R^{3x3}$ is the robot’s rotation. Because the rotation matrix is not additive, it is not convenient for iterative optimization; thus, we use Euler angles and Lie algebra to represent the robot’s rotation in the following derivation. Euler angles easily distinguish the constraint dimension of each factor, and the pitch does not reach ±90 degrees during the movement of TWIP, avoiding any problems with gimbal lock. In this paper, $\theta \;=\;{\left[\begin{array}{ccc} \theta_{r} & \theta_{p} & \theta_{y} \end{array}\right]}^{T}$ represents the Euler angle and $\theta_{r},\;\theta_{p},\;\theta_{y}$ represents the values of the roll, pitch, and yaw angles, respectively. The rotation integral based on Euler angles is only exact up to the first order in robot state estimation [26]. Therefore, we use Lie algebra, $\phi \;=\;{\left[\begin{array}{ccc} \phi_{1} & \phi_{2} & \phi_{3} \end{array}\right]}^{T}$, to represent the rotation in the optimization process. Lie algebra can be converted to rotation matrices using Equation (Equation 7). According to the motion characteristics of TWIP, $\left[\begin{array}{cccc} x & y & \theta_{p} & \theta_{y} \end{array}\right]$ is the result of the robot’s motion, which is named the primary parameter, while $\left[\begin{array}{cc} z & \theta_{r} \end{array}\right]$ is the motion disturbance caused by uneven ground or the vibration of the robot, and is named the second parameter. The partial parameters in the figure are shown in Table 1.

### 4.2. Coupling of Lidar–IMU–Odometer via Factor Graph

In this section, we introduce five types of factors to construct factor graphs in order to estimate the motion state of TWIP. The robot’s state at a specific time is attributed to the graph’s nodes. According to the constraints of factors on nodes, these factors are divided into holonomic and nonholonomic constraints. The holonomic constraint factors include the lidar pose factor, the IMU pre-integration factor, and the loop closure factor. The nonholonomic constraint factor consists of the odometer pre-integration and ground constraints. A new robot state node is added to the graph when the change in robot pose exceeds a user-defined threshold. The factor graph is optimized upon the insertion of a new node using incremental smoothing and mapping with the Bayes tree (iSAM2) [17]. The process for generating these factors is described in the following sections.

#### 4.2.1. Lidar Pose Factor

Mechanical 3D LiDAR senses its surroundings by rotating a vertically aligned array of laser beams. When a lidar keyframe $F_{i}$ is selected, we first de-skew the point cloud. The movement of the carrier causes point cloud distortion during the lidar data collection process, and the points in a lidar scan are not collected at the same time; that is, the frames of different lidar points are inconsistent. The bias of the IMU obtained by the factor graph optimization is used to correct the IMU measurement, which can estimate the pose of the carrier. Here, we find the pose corresponding to each lidar point, transform them into the same coordinate system, and obtain the resulting point cloud without distortion. The feature point matching method is used in this paper because it is more robust and efficient than noise-sensitive matching methods, such as Iterative Closest Point (ICP). Similar to [7], this paper uses plane feature points and edge feature points as matching points, which are extracted by smoothness. For any point $p_{k}$ in $F_{i}$, We find ten straight points of $p_{k}$ from the same sweep, of which half are on either side of $p_{k}$. The smoothness of $p_{k}$ can then be calculated by the following formula:(18)$$c=\frac{1}{\left|D\right|\cdot ∥r_{k}∥}∥\sum_{j\in D,j\neq k}\left(r_{j}-r_{k}\right)∥$$
where *D* is the point set adjacent to this point we selected and $\left|D\right|$ is the number of points in *D*.

We use $F_{i}=\left\{F_{i}^{\varepsilon },F_{i}^{\pi }\right\}$ to represent the feature points extracted from the lidar scan at time *i*, where $F_{i}^{\pi }$ represents the planar points and $F_{i}^{\varepsilon }$ represents the edge points. Points with slight smoothness are selected as plane points and points with significant smoothness are selected as edge points. A detailed description of the feature extraction process is provided in [7].

After the above processing is complete, we have the feature points of the current lidar frame without distortion. We then need to find the plane and edge line corresponding to the current frame feature point in the target point cloud for the feature-based matching method. Unlike traditional algorithms such as lio_sam and lego_loam, which take lidar scans close to the current frame in the time series as matching point clouds, this paper searches for the matching point cloud of the current frame in Euclidean space, which is more accurate because our optimization process is to minimize the point-to-line and point-to-plane spatial distances. Therefore, in this algorithm the position of the current robot is taken as a priori, and a part of the point cloud is extracted from the global map to align with the feature points without distortion in the current frame in order to estimate the pose of the robot. The IMU provides the initial position. The global map consists of edge feature maps and plane feature maps, which are updated and maintained, respectively. The global map is stored in an incremental 3D tree [18]. Compared with the traditional dynamic data structure, incremental 3D tree has an efficient nearest neighbor search and supports the incremental update of the map. Similar to [18], we convert the current frame to the world frame to obtain ${}^{W}F_{i}=\left\{{}^{W}F_{i}^{\varepsilon },{}^{W}F_{i}^{\pi }\right\}$, while the IMU provides the initial pose. We find the *N* nearest edge points in the global edge map for any edge feature point $p_{i}^{\varepsilon }$, then calculate the variance of these nearby points. If one of the eigenvalues of the variance matrix is significantly larger than the other two, the points are distributed in a line. The eigenvector $n_{i}^{\varepsilon }$, which corresponds to the largest eigenvalue, is the direction vector of the line. The edge point $p_{j}^{\varepsilon }$ closest to the feature point is taken as a point on the line. The distance from the feature point to the global line is
(19)$$d_{i}^{\varepsilon }={∥{\left(n_{i}^{\varepsilon }\right)}^{\wedge }\left({\tilde{T}}_{L_{i}}^{W}p_{i}^{\varepsilon }-p_{j}^{\varepsilon }\right)∥}_{2}$$

For any plane feature point $p_{i}^{\pi }$, we find the *M* nearest plane points in the global plane map and calculate the variance of these nearby points. If one of the eigenvalues of the variance matrix is significantly smaller than the other two, the points are distributed in a plane. The eigenvector $n_{i}^{\pi }$, which corresponds to the smallest eigenvalue, is the normal vector of the plane. We select the closest plane feature point $p_{j}^{\pi }$ to the feature point as a point on the plane. Then, the distance from the feature point to the global plane is
(20)$$d_{i}^{\pi }=\left|\left(n_{i}^{\pi }\right)\cdot \left({\tilde{T}}_{L_{i}}^{W}p_{i}^{\pi }-p_{j}^{\pi }\right)\right|$$

We can obtain the optimal pose of the current lidar scan relative to the world frame by minimizing the point-to-line and point-to-face distances:(21)$$\min_{{\tilde{T}}_{L_{i}}^{W}}\sum_{i=0}^{N}{\left(d_{i}^{\varepsilon }\right)}^{2}+\sum_{i=0}^{M}{\left(d_{i}^{\pi }\right)}^{2}$$

After optimization, we have the optimal absolute transformation ${\tilde{T}}_{L_{i}}^{W}$ from the current keyframe to the world frame, which is used to measure the lidar pose factor.

We obtain the residual of the lidar pose factor using the following formula:(22)$$r_{L_{i}}=\log\left({\left(T_{B_{i}}^{W}T_{L}^{B}\right)}^{T}{\tilde{T}}_{L_{i}}^{W}\right)$$
where $T_{L}^{B}$ is the transformation between the lidar coordinate system and the robot body coordinate system, calibrated in advance.

The variance of the factor is calculated by the method used in [26]:(23)$$\Sigma_{L_{i}}=\sum_{k=1}^{N}H_{k}^{T}\Lambda^{-1}H_{k}$$
(24)$$H_{k}=\left[\begin{array}{cc} {\left({\tilde{R}}_{L_{i}}^{W}\;p_{k}+{\tilde{P}}_{L_{i}}^{W}\right)}^{\wedge } & -{\tilde{R}}_{L_{i}}^{W} \end{array}\right]$$
(25)$$\Lambda =\left[\begin{array}{ccc} 2\sigma_{x}^{2} & 0 & 0\\ 0 & 2\sigma_{y}^{2} & 0\\ 0 & 0 & 2\sigma_{z}^{2} \end{array}\right]$$
where $p_{k}$ is the feature point in the current lidar keyframe, and is transformed into the matching point cloud through the optimal transformation, ${\tilde{T}}_{L_{i}}^{W}=\left[\begin{array}{cc} {\tilde{R}}_{L_{i}}^{W} & {\tilde{P}}_{L_{i}}^{W}\\ 0 & 1 \end{array}\right]$, obtained through the above matching. Here, *N* is the number of features in the current lidar keyframe, Λ is the zero-mean white Gaussian noise of the measurements, and $\sigma_{x},\sigma_{y}$ and $\sigma_{z}$ are the noise sigma (in meters) on the *x*, *y*, and *z* axes, respectively. The sigmas are multiplied by two, as both the target point and the matching are affected by this noise.

#### 4.2.2. IMU Pre-Integration Factor

The main components of an IMU are a three-axis accelerometer and a three-axis gyroscope that measure the linear acceleration and angular velocity of the carrier, respectively. The pose transformation of the carrier to an inertial frame can be obtained by integral operation. The raw measurements, ${\tilde{a}}_{t}$ and ${\tilde{\omega }}_{t}$, at moment *t* are provided by
(26)$$\begin{array}{cc} \; & {\tilde{a}}_{t}=a_{t}+b_{a_{t}}+R_{W}^{I_{t}}g^{w}+n_{a}\\ \; & {\tilde{\omega }}_{t}=\omega_{t}+b_{\omega_{t}}+n_{\omega } \end{array}$$

These IMU measurements are measured in the IMU frame, and are affected by the additive noise, $n_{a}$,$n_{\omega }$, the acceleration bias, $b_{a_{t}}$, and the gyroscope bias, $b_{\omega_{t}}$. Generally, in this paper we model the bias as a random walk with Gaussian derivatives, while the additive noise of the accelerometer and the gyroscope is assumed to be subject to a Gaussian distribution:(27)$$n_{a}∼N\left(0,\sigma_{a}^{2}\right),n_{\omega }∼N\left(0,\sigma_{\omega }^{2}\right)$$

Over time, high-frequency IMU measurement leads to rapid growth in the number of variables in the optimization, increasing the computational cost. We therefore convert the IMU measurements between keyframes into motion constraints via the pre-integration technique, reducing the computational cost and providing a factor, namely, the IMU pre-integration factor. The parameters to be optimized include the robot’s pose and the bias of the IMU.

According to the kinematic model in [26], the pose and velocity of the current keyframe in the world frame can be obtained from the measurement of the previous frame. With the pose and velocity of a keyframe *i*, the pose and velocity of the current keyframe *j* are:(28)$$\begin{array}{cc} P_{I_{j}}^{W}= & P_{I_{i}}^{W}+v_{I_{i}}^{W}\Delta t+{\;\int \int \;}_{t\in [i,j]}\left(R_{I_{t}}^{W}\left({\tilde{a}}_{t}-b_{a_{t}}-n_{a}\right)-g^{W}\right)dt^{2}\\ v_{I_{j}}^{W}= & v_{I_{i}}^{W}+\int_{t\in [i,j]}\left(R_{I_{t}}^{W}\left({\tilde{a}}_{t}-b_{a_{t}}-n_{a}\right)-g^{W}\right)dt\\ R_{I_{j}}^{W}= & R_{I_{i}}^{W}\mathrm{Exp}\left(\int_{t\in [i,j]}\left({\tilde{\omega }}_{t}-b_{\omega_{t}}-n_{\omega }\right)dt\right) \end{array}$$

Using Formula (28), we iterate the IMU integral between times *i* and *j*, and obtain:(29)$$\begin{array}{cc} \; & I_{I_{j}}^{W}=P_{I_{i}}^{W}+v_{I_{i}}^{W}\Delta t+\sum_{k=i}^{j-1}\left(v_{I_{k}}^{W}\Delta t-\frac{1}{2}g^{W}\Delta t^{2}+\frac{1}{2}R_{I_{k}}^{W}\left({\tilde{a}}_{k}-b_{a_{k}}-n_{a}\right)\Delta t^{2}\right)\\ \; & v_{I_{j}}^{W}=v_{I_{i}}^{W}+g^{W}\Delta t_{ij}+\sum_{k=i}^{j-1}R_{I_{k}}^{W}\left({\tilde{a}}_{k}-b_{a_{k}}-n_{a}\right)\Delta t\\ \; & R_{I_{j}}^{W}=R_{I_{i}}^{W}\prod_{k=i}^{j-1}\mathrm{Exp}\left({\tilde{\omega }}_{k}-b_{\omega_{k}}-n_{\omega }\right)\Delta t \end{array}$$
where Δt is the interval between two adjacent measurements, $\Delta t_{ij}=\sum_{k=i}^{j-1}\Delta t$. Next, we separate the parts related to IMU measurement from the pose and velocity. The parts related to IMU measurement are as follows:(30)$$\begin{array}{cc} \; & \Delta {\tilde{P}}_{I_{j}}^{I_{i}}=\sum_{k=i}^{j-1}\left[\Delta V_{k}^{i}\Delta t+\frac{1}{2}R\left(\gamma_{k}^{i}\right)\left({\tilde{a}}_{k}-b_{a_{k}}\right)\Delta t^{2}\right]-\delta P_{ij}\\ \; & \Delta {\tilde{V}}_{I_{j}}^{I_{i}}=\sum_{k=i}^{j-1}R_{k}^{i}\left({\tilde{a}}_{k}-b_{a_{k}}\right)\Delta t-\delta V_{ij}\\ \; & \Delta {\tilde{R}}_{I_{j}}^{I_{i}}=\left(\prod_{k=i}^{j-1}\mathrm{Exp}\left(\left({\tilde{\omega }}_{k}-b_{\omega_{k}}\right)\Delta t\right)\right)\mathrm{Exp}\left(-\delta \varphi_{ij}\right) \end{array}$$
where $\delta P_{ij}$, $\delta V_{ij}$, and $\delta \varphi_{ij}$ are the measurement error of IMU, and are obtained by iterating the initial value at time *i*. In order to avoid repeated integration due to changes in the bias during the optimization process, we use first-order approximation to linearize the bias terms, as follows:(31)$$\begin{array}{c} {\Delta \tilde{P}}_{I_{j}}^{I_{i}}\approx {\Delta \tilde{P}}_{I_{j}}^{I_{i}}+J_{b_{a}}^{\Delta \tilde{P}}\delta b_{a}+J_{b_{w}}^{\Delta P}\delta b_{w}\\ {\Delta \tilde{V}}_{I_{j}}^{I_{i}}\approx {\Delta \tilde{V}}_{I_{j}}^{I_{i}}+J_{b_{a}}^{\Delta \tilde{V}}\delta b_{a}+J_{b_{w}}^{\Delta V}\delta b_{w}\\ {\Delta \tilde{R}}_{I_{j}}^{I_{i}}\approx {\Delta \tilde{R}}_{I_{j}}^{I_{i}}\mathrm{Exp}\left(J_{b_{\omega }}^{\Delta \tilde{R}}\delta b_{w}\right) \end{array}$$

The items related to the state at times *i* and *j* are as follows:(32)$$\begin{array}{c} \Delta P_{I_{j}}^{I_{i}}={\left(R_{B_{i}}^{W}R_{I}^{B}\right)}^{T}\left(P_{I_{j}}^{W}-P_{I_{i}}^{W}-v_{I_{i}}^{W}\Delta t+\frac{1}{2}g^{W}\Delta t^{2}\right)\\ \Delta V_{I_{j}}^{I_{i}}={\left(R_{B_{i}}^{W}R_{I}^{B}\right)}^{T}\left(v_{I_{j}}^{W}-v_{I_{i}}^{W}+g^{W}\Delta t\right)\\ \Delta R_{I_{j}}^{I_{i}}={\left(R_{B_{i}}^{W}R_{I}^{B}\right)}^{T}R_{B_{j}}^{W}R_{I}^{B} \end{array}$$

We construct the motion constraints of the IMU between two keyframes, which naturally provide the IMU pre-integration factor. This factor takes the bias of the IMU as an optimization parameter, and its residual is defined as follows:(33)$$\begin{array}{cc} r_{P} & =\Delta P_{I_{j}}^{I_{i}}-\Delta {\tilde{P}}_{I_{j}}^{I_{i}}\\ \; & ={\left(R_{B_{i}}^{W}R_{I}^{B}\right)}^{T}\left(P_{B_{j}}^{W}-P_{B_{i}}^{W}-v_{I_{i}}^{W}\Delta t+\frac{1}{2}g^{W}\Delta t^{2}\right)-\sum_{k=i}^{j-1}\left[\Delta V_{k}^{i}\Delta t+\frac{1}{2}R\left(\gamma_{k}^{i}\right)\left({\tilde{a}}_{k}-b_{a_{k}}\right)\Delta t^{2}\right]\\ r_{v} & =\Delta V_{I_{j}}^{I_{i}}-\Delta {\tilde{V}}_{I_{j}}^{I_{i}}\\ \; & ={\left(R_{B_{i}}^{W}R_{I}^{B}\right)}^{T}\left(v_{I_{j}}^{W}-v_{I_{i}}^{W}+g^{W}\Delta t\right)-\sum_{k=i}^{j-1}R_{k}^{i}\left({\tilde{a}}_{k}-b_{a_{k}}\right)\Delta t\\ r_{R} & =\log\left({\left(\Delta R_{I_{j}}^{I_{i}}\right)}^{T}\Delta {\tilde{R}}_{I_{j}}^{I_{i}}\right)\\ \; & =\log\left({\left(R_{B_{j}}^{W}R_{I}^{B}\right)}^{T}R_{B_{i}}^{W}R_{I}^{B}\prod_{k=i}^{j-1}\mathrm{Exp}\left(\left({\tilde{\omega }}_{k}-b_{w_{t_{i}}}\right)\Delta t\right)\right)\\ r_{b_{a}} & =b_{a_{j}}-b_{a_{i}}\\ r_{b_{\omega }} & =b_{\omega_{j}}-b_{\omega_{i}} \end{array}$$
where $R_{I}^{B}$ is the rotation between the lidar coordinate system and the robot body coordinate system, calibrated in advance.

The variance of the IMU pre-integration factor can be obtained iteratively from an initial value 0 using the method in [26].

#### 4.2.3. Odometry Pre-Integration Factor

Generally, wheeled robots are equipped with wheel encoders that often provide noisy and only intermittently reliable measurements of the motion of each wheel. In order to reduce the amount of computation and accumulated error, by analogy to IMU we use the pre-integration technique on SE(2) to derive the constraints of the odometer. Unlike general ground mobile robots, TWIP swings back and forth in addition to plane motion, while the encoder only provides two-dimensional plane motion. In order to analyze the constraints of the encoder more conveniently, we add an auxiliary coordinate system, that is, a plane coordinate system. The coordinate system is fixed to the robot; its origin coincides with the origin of the robot body frame, the *x**y* plane coincides with the physical plane, the *z*-axis is vertically upward, and the *y* axis coincides with the y axis of the robot body frame, as shown in Figure 6. The encoder measurements between two adjacent keyframes indicate the pose transformation between the plane coordinate systems of the corresponding moments. Therefore, we analyze the constraints provided by the encoder in the planar coordinate system.

The measurement data of the encoder can be converted into linear velocity and angular velocity on a two-dimensional plane:(34)$$\begin{array}{c} v=\frac{r_{l}\omega_{l}+r_{r}\omega_{r}}{2}\\ \omega =\frac{r_{l}\omega_{l}-r_{r}\omega_{r}}{l} \end{array}$$
where $\omega_{l}$ and $\omega_{r}$ are the rotational velocities of the left and right wheels, respectively, $r_{l}$ and $r_{r}$ are their corresponding radii, and *l* denotes the robot’s baseline.

We model the encoder’s measurement error with a Gaussian model to process the noisy odometer data. We begin by deriving the measurement model, assuming that between successive odometer readings the motion is planar; the encoder provides the following measurements in the planar coordinate system of the current keyframe:(35)$$\begin{array}{c} {\tilde{v}}_{i}=v_{i}+\varepsilon_{vi}\\ {\tilde{w}}_{i}=\omega_{i}+\varepsilon_{\omega i} \end{array}$$
where $v_{i}=\left[\begin{array}{c} v_{i}\\ 0 \end{array}\right],\varepsilon_{vi}=\left[\begin{array}{c} \varepsilon_{vi}\\ 0 \end{array}\right]$; $\varepsilon_{vi}$ and $\varepsilon_{wi}$ represent zero-mean Gaussian noise.

With two consecutive lidar keyframes, *i* and *j*, we obtain the pose of the plane coordinate system corresponding to the current keyframe in the world coordinate system from the pose of the previous keyframe and the measurements of the encoder:(36)$$\begin{array}{c} \phi_{\pi_{j}}^{W}=\phi_{\pi_{i}}^{W}+\sum_{k=i}^{j-1}\left({\tilde{w}}_{k}-\varepsilon_{\omega k}\right)\Delta t\\ P_{\pi_{j}}^{W}=P_{\pi_{i}}^{W}+\sum_{k=i}^{j-1}R\left(\phi_{\pi_{k}}^{W}\right)\left(\left({\tilde{v}}_{k}-\varepsilon_{vk}\right)\Delta t\right) \end{array}$$
where $\phi_{\pi_{k}}^{W},k=i,j$ is the yaw angle of the plane coordinate system corresponding to keyframe *k* relative to the world coordinate system and $P_{\pi_{k}}^{W}=\left[\begin{array}{c} x_{k}\\ y_{k} \end{array}\right],k=i,j$ is the 2D position of the plane coordinate system corresponding to keyframe *k* relative to the world coordinate system:(37)$$R\left(\phi \right)=\exp\left(\phi^{\wedge }\right)=\exp\left(\left[\begin{array}{cc} 0 & -\phi \\ \phi & 0 \end{array}\right]\right)=\left[\begin{array}{cc} \cos\phi & -\sin\phi \\ \sin\phi & \cos\phi \end{array}\right]$$

Note that the position propagated here depends on the rotational state. In order to generate a measurement between two keyframes *i* and *j* without depending on the robot’s state in the world coordinate system, we place the items related to the measurement together and eliminate the items related to the state of *i* and *j*. Equation (Equation 36) can be written as:(38)$$\begin{array}{c} \Delta \phi_{ij}=\phi_{\pi_{j}}^{W}-\phi_{\pi_{i}}^{W}\\ \;\;\;\;\;\;\;\;\;=\sum_{k=i}^{j-1}\left({\tilde{\omega }}_{i}-\varepsilon_{\omega }\right)\Delta t\\ \;\;\;\;\;\;\;\;\;=\sum_{k=i}^{j-1}{\tilde{\omega }}_{i}\Delta t-\sum_{k=i}^{j-1}\varepsilon_{\omega k}\Delta t\\ \;\;\;\;\;\;\;\;\;=\Delta {\tilde{\phi }}_{ij}-\delta \phi_{ij} \end{array}$$
(39)$$\begin{array}{c} \Delta P_{ij}=R\left(-\phi_{\pi_{i}}^{W}\right)\left(P_{\pi_{j}}^{W}-P_{\pi_{i}}^{W}\right)\\ \;\;\;\;\;\;\;\;\;\;=\sum_{k=i}^{j-1}R\left(\phi_{ik}\right)\left(\left({\tilde{v}}_{k}-\varepsilon_{vk}\right)\Delta t\right)\\ \;\;\;\;\;\;\;\;\;\;\approx \sum_{k=i}^{j-1}R\left({\tilde{\phi }}_{ik}\right)\exp\left(-\delta \phi_{ik}\right)\left(\left({\tilde{v}}_{k}-\varepsilon_{vk}\right)\Delta t\right)\\ \;\;\;\;\;\;\;\;\;\;\approx \sum_{k=i}^{j-1}R\left({\tilde{\phi }}_{ik}\right)\left(I_{2\times 2}-\delta \phi_{ik}^{\wedge }\right)\left(\left({\tilde{v}}_{k}-\varepsilon_{vk}\right)\Delta t\right)\\ \;\;\;\;\;\;\;\;\;\;\approx \sum_{k=i}^{j-1}R\left({\tilde{\phi }}_{ik}\right){\tilde{v}}_{k}\Delta t-\sum_{k=i}^{j-1}R\left({\tilde{\phi }}_{ik}\right)\varepsilon_{vk}\Delta t-\sum_{k=i}^{j-1}R\left({\tilde{\phi }}_{ik}\right)1^{\wedge }{\tilde{v}}_{k}\Delta t\delta \phi_{ik}\\ \;\;\;\;\;\;\;\;\;\;=\Delta {\tilde{P}}_{ij}-\delta P_{ij} \end{array}$$

In order to obtain the error transfer model, we derive the integrated noise terms in the form of iterative transfer:(40)$$\begin{array}{c} \delta \phi_{i,k+1}=\delta \phi_{ik}+\varepsilon_{\omega k}\Delta t\\ \delta p_{i,k+1}=\delta p_{ik}+R\left({\tilde{\phi }}_{ik}\right)1^{\wedge }{\tilde{v}}_{k}\Delta t\delta \phi_{ik}+R\left({\tilde{\phi }}_{ik}\right)\Delta t\varepsilon_{vk} \end{array}$$

Writing Equation (Equation 40) in a compact form, we then have:(41)$${\left[\begin{array}{c} \delta p_{i,k+1}\\ \delta \phi_{i,k+1} \end{array}\right]}_{3\times 1}=A_{k}{\left[\begin{array}{c} \delta p_{ik}\\ \delta \phi_{ik} \end{array}\right]}_{3\times 1}+B_{k}{\left[\begin{array}{c} \varepsilon_{vk}\\ \varepsilon_{\omega k} \end{array}\right]}_{3\times 1}$$
where
(42)$$\begin{array}{c} A_{k}=\left[\begin{array}{cc} I_{2\times 2} & R\left(\phi_{ik}\right)1^{\wedge }{\tilde{v}}_{k}\Delta t\\ 0 & 1 \end{array}\right] \end{array}$$
(43)$$\begin{array}{c} B_{k}=\left[\begin{array}{cc} R\left(\phi_{ik}\right)\Delta t & 0_{2\times 1}\\ 0_{1\times 2} & \Delta t \end{array}\right] \end{array}$$

Iterating from the initial value, $\Sigma_{\delta i}=0_{3\times 3}$, at the time *i*, we can incrementally calculate the pre-integrated terms $\Delta \phi_{ij}$ and $\Delta P_{ij}$ and their covariance matrix, $\Sigma_{\delta }=\left[\begin{array}{cc} \Sigma_{\delta P_{ij}} & 0_{2\times 1}\\ 0_{1\times 2} & \Sigma_{\delta \phi_{ij}} \end{array}\right]$.

Thus far, we have the encoder’s pre-integration terms, $\Delta {\tilde{\phi }}_{ij}$ and $\Delta {\tilde{P}}_{ij}$, which provide the measurements of the odometer constraints. Next, we formulate the residual of the odometer pre-integration constraint. Through the definition of the plane coordinate system, we can obtain the rotation of the plane coordinate system corresponding to the keyframes *i* and *j* relative to the world coordinate system, respectively:(44)$$\begin{array}{c} R_{\pi_{i}}^{W}=\left[\begin{array}{ccc} -e_{3}\times R_{B_{i}}^{W}e_{2} & R_{B_{i}}^{W}e_{2} & e_{3} \end{array}\right] \end{array}$$
(45)$$\begin{array}{c} R_{\pi_{j}}^{W}=\left[\begin{array}{ccc} -e_{3}\times R_{B_{j}}^{W}e_{2} & R_{B_{j}}^{W}e_{2} & e_{3} \end{array}\right] \end{array}$$
where $e_{i}$ is the ith column vector of the identity matrix, $I_{3\times 3}$.

Then, the relative transformation of the plane coordinate system corresponding to keyframes *i* and *j* is as follows:(46)$$\begin{array}{c} R_{\pi_{i}}^{\pi_{j}}={\left(R_{\pi_{j}}^{W}\right)}^{T}R_{\pi_{i}}^{W}\\ \;\;\;\;\;\;=\left[\begin{array}{c} {\left(-e_{3}\times R_{B_{j}}^{W}e_{2}\right)}^{T}\\ {\left(R_{B_{j}}^{W}e_{2}\right)}^{T}\\ e_{3}^{T} \end{array}\right]\left[\begin{array}{ccc} -e_{3}\times R_{B_{i}}^{W}e_{2} & R_{B_{i}}^{W}e_{2} & e_{3} \end{array}\right]\\ \;\;\;\;\;\;=\left[\begin{array}{ccc} -e_{2}^{T}{\left(R_{B_{j}}^{W}\right)}^{T}e_{3}^{\wedge }e_{3}^{\wedge }R_{B_{i}}^{W}e_{2} & e_{2}^{T}{\left(R_{B_{j}}^{W}\right)}^{T}e_{3}^{\wedge }R_{B_{i}}^{W}e_{2} & 0\\ -e_{2}^{T}{\left(R_{B_{j}}^{W}\right)}^{T}e_{3}^{\wedge }R_{i}^{W}e_{2} & e_{2}^{T}{\left(R_{B_{j}}^{W}\right)}^{T}R_{B_{i}}^{W}e_{2} & 0\\ 0 & 0 & 1 \end{array}\right] \end{array}$$

As can be seen from Appendix A and Appendix B, the measurement of the rotating part of the encoder $\Delta {\tilde{\phi }}_{ij}$ provides only one constraint, and the residual of the rotation part is:(47)$$r_{R}=e_{2}^{T}{\left(R_{B_{j}}^{W}\right)}^{T}R_{B_{i}}^{W}e_{2}-\cos\left(\Delta {\tilde{\phi }}_{ij}\right)$$

We obtain the variance of the rotation residuals using a first-order linear approximation of the trigonometric functions:(48)$$\begin{array}{c} \cos\phi \approx \cos\left(\Delta \phi_{ij}\right)-\sin\left(\Delta \phi_{ij}\right)\left(\phi -\Delta \phi_{ij}\right) \end{array}$$
(49)$$\begin{array}{c} \Sigma_{r_{R}}=\sin^{2}\left(\Delta \phi_{ij}\right)\Sigma_{\delta \phi_{ij}} \end{array}$$

Then, its Jacobian matrix relative to the TWIP motion state (the pose of the robot body frame relative to the world frame) is
(50)$$\begin{array}{c} J_{\phi_{i}}^{r_{R}}=-e_{2}^{T}{\left(R_{B_{j}}^{W}\right)}^{T}R_{B_{i}}^{W}e_{2}^{\wedge } \end{array}$$
(51)$$\begin{array}{c} J_{\phi_{j}}^{r_{R}}=e_{2}^{T}{\left({\left(R_{B_{j}}^{W}\right)}^{T}R_{B_{i}}^{W}e_{2}\right)}^{\wedge } \end{array}$$
and the residual of the translation part is
(52)$$\begin{array}{c} r_{P}={\left[{\left(R_{\pi_{i}}^{W}\right)}^{T}\left(P_{B_{j}}^{W}-P_{B_{i}}^{W}\right)\right]}_{xy}-\Delta {\tilde{P}}_{ij}\\ \;\;\;\;\;\;={\left[\begin{array}{c} {\left(-e_{3}\times R_{B_{i}}^{W}e_{2}\right)}^{T}\left(P_{B_{j}}^{W}-P_{B_{i}}^{W}\right)\\ {\left(R_{B_{i}}^{W}e_{2}\right)}^{T}\left(P_{B_{j}}^{W}-P_{B_{i}}^{W}\right)\\ 0 \end{array}\right]}_{xy}-\Delta {\tilde{P}}_{ij} \end{array}$$
where ${\left[\right]}_{xy}$ represents the first two rows of the vector, and its covariance is
(53)$$\Sigma_{r_{P}}=\Sigma_{\delta P_{ij}}$$

Then, its Jacobian matrix relative to the motion state of TWIP is
(54)$$\begin{array}{c} J_{\phi_{i}}^{r_{P}}=\left[\begin{array}{c} e_{2}^{T}{\left({\left(R_{B_{i}}^{W}\right)}^{T}e_{3}^{\wedge }\left(P_{B_{j}}^{w}-P_{B_{i}}^{w}\right)\right)}^{\wedge }\\ e_{2}^{T}{\left({\left(R_{B_{i}}^{W}\right)}^{T}\left(P_{B_{j}}^{w}-P_{B_{i}}^{w}\right)\right)}^{\wedge } \end{array}\right] \end{array}$$
(55)$$\begin{array}{c} J_{P_{B_{i}}^{W}}^{r_{P}}=\left[\begin{array}{c} -e_{2}^{T}{\left(R_{B_{i}}^{W}\right)}^{T}e_{3}^{\wedge }\\ -e_{2}^{T}{\left(R_{B_{i}}^{W}\right)}^{T} \end{array}\right] \end{array}$$
(56)$$\begin{array}{c} J_{P_{B_{j}}^{W}}^{r_{P}}=\left[\begin{array}{c} e_{2}^{T}{\left(R_{B_{i}}^{W}\right)}^{T}e_{3}^{\wedge }\\ e_{2}^{T}{\left(R_{B_{i}}^{W}\right)}^{T} \end{array}\right] \end{array}$$

#### 4.2.4. Ground Constraint Factor

This paper proposes a new nonholonomic constraint factor in order to add ground constraints. We assume that TWIP moves on an approximately flat surface and that the robot is always constrained by the ground plane during movement. Ideally, the *z* and pitch angle would remain constant during the planar motion for a two-wheeled inverted pendulum robot. In the natural environment, however, there are motion disturbances in the other dimensions of the robot’s pose due to rough terrain and shaking in the robot’s motion. In order to reduce the error caused by this motion disturbance, we use a random model to introduce ground constraints and assume that the motion disturbance is a zero-mean Gaussian model. This is more in line with the natural environment, allowing a least squares to be used to model the ground constraints, which naturally adds the ground constraints as a new factor to the factor graph. As shown in Figure 7, when TWIP moves on an ideal plane the roll angle and *z* of its body frame relative to the world frame should both be zero. Therefore, the residual of the ground constraint is defined as follows:(57)$$\begin{array}{c} r_{R}=e_{3}^{T}R_{B_{i}}^{W}e_{2} \end{array}$$
(58)$$\begin{array}{c} r_{P}=e_{3}^{T}\cdot P_{B_{i}}^{W} \end{array}$$

The corresponding covariance matrix is
(59)$$\Sigma_{r_{p}}=diag\left(\begin{array}{cc} \sigma_{r}^{2} & \sigma_{z}^{2} \end{array}\right)$$

In this paper, we assume that the motion disturbance of the robot due to rough terrains or motion shaking obeys the Gaussian distribution; $\sigma_{z}$ and $\sigma_{r}$ are the noise sigma of the *z* and roll, respectively, which are determined by the conditions of the terrain and the robot’s structure.

The Jacobian matrix relative to the state of TWIP is
(60)$$\begin{array}{c} J_{\phi_{i}}^{r_{R}}=-e_{3}^{T}R_{B_{i}}^{W}e_{2}^{\wedge } \end{array}$$
(61)$$\begin{array}{c} J_{P_{B_{i}}^{W}}^{r_{P}}=e_{3}^{T} \end{array}$$

#### 4.2.5. Loop Closure Factor

When a new lidar keyframe, $F_{i+1}$, arrives, its corresponding robot state, $K_{i+1}$, is added to the factor graph. We obtain the absolute pose of the current lidar keyframe as in Section 4.2.1. We obtain the absolute pose of the current lidar keyframe as in Section 4.2.1, and use this pose as a prior to find the previous keyframe closest to $F_{i+1}$ in Euclidean space as a candidate frame. For example, the lidar keyframe $F_{3}$ in Figure 3 is one of the returned loop closure candidates. Then, taking the position of the lidar keyframe $F_{3}$ as the center, we extract the point cloud in the cuboid with a length of 10 m and a height of 5 m in the global map and transform the point cloud into the robot body coordinate system corresponding to the lidar keyframe $F_{3}$ to construct a local map $M_{i+1}$. We try to match $F_{i+1}$ to the local map $M_{i+1}$ using the scan-matching method discussed in Section 4.2.1. In the local map, we select the *N* nearest feature points for each feature point of the current frame to fit a line or plane. After optimization, we obtain the relative transformation, $\Delta {\tilde{T}}_{3}^{i+1}$, and add it to the graph as a loop closure factor. Throughout this paper, we choose to set the loop closure’s search distance to 15 m from the new lidar keyframe.

## 5. Experiment

We conducted experiments on our collected datasets, as we could not find an open dataset to provide calibrated Lidar–IMU–Odometry for a two-wheeled inverted pendulum robot. All experiments were carried out with an Intel CPU i7-8559U (8 cores@2.70 GHz) laptop computer with 8 GB RAM. As shown in Figure 8, the platform was equipped with a Velodyne VLP-16 lidar oprating at 10 HZ, an XSENS MTI-G-710 IMU to provide acceleration and angular velocity at 200 HZ, and an encoder with 1khz, all of which were synchronized and calibrated in advance.

The dataset comes from two scenes; one is a long straight corridor scene, named Dataset Corridor, and the other is a flat outdoor road, named Dataset Outdoor. The environment of the two datasets is shown in Figure 9. As with the process of data collection, the actual ground truth of the robot is unavailable. Inspired by [25], we equipped TWIP with downward-viewing cameras on the front and rear (their position is shown in Figure 7) and an attitude sensor with a dynamic accuracy of $0.2^{\circ }$. Tens of Aruco code markers were laid out on the floor, and their locations in a predefined world reference were measured with a location error of less than 1 cm. Transformations between industrial cameras and attitude sensors and the robot were all pre-calibrated. The two industrial cameras detected and recorded these marks during the robot’s movement. In order not to affect the algorithm in this paper as well as to improve the true accuracy, the trajectory truth was obtained offline. We extracted the marked corner points in each recorded image and approximated the trajectory truth by optimizing the reprojection error of the image and the pose constraints of the pose sensor in an offline batch. In order to verify the algorithm used in this article, we tested the constructed SLAM algorithm and the state-of-the-art lidar slam method (Fast_lio2 [16], Lio_sam [18], Loam [7]) on the same dataset.

### 5.1. Experimental Results and Analysis of Indoor Corridors

Figure 10 shows the results of our algorithm for localization and mapping with Dataset Corridor. Figure 11 compares trajectories in the xy direction recovered by our algorithm, Fast_lio2, Lio_sam, Loam, Odometry, and ground truth on the Long Straight Corridor dataset. The odometer is a trajectory obtained using the less accurate odometer raw data calculated directly from the encoder to emphasize the correction effect of the lidar method on the odometer’s accumulated error. Meanwhile, to compare the long-term positioning accuracy of the algorithms, all algorithms did not add a loop detection module, and the pose estimation of all algorithms was performed in the robot body frame. Figure 12 and Figure 13 show the corresponding trajectory errors relative to the ground truth, respectively. It can be seen from Figure 12 that the *x*-direction is the running direction of TWIP, which has fewer constraints, and these algorithms have more significant errors in the *x*-direction. Among them, the accuracy of loam in the *x*-direction is significantly lower than the other algorithms due to the lack of measurements from other sensors. There is a constraint from the wall in the *y*-direction; all the lidar-based SLAM algorithms have little difference in accuracy, as they all use the same matching method. While the results by our method exhibit much better accuracy, there is a smaller upper limit of errors in the *x* and *y* directions. Attention should be paid to the *z*-axis values in Figure 14, where the *z*-coordinates estimated by our method are well-bounded around zero (which is consistent with natural indoor environments) while the other algorithms have significantly larger upper bounds. Using a nonholonomic constraint factor and adding planar constraints and encoder pre-integration constraints improves indoor pose estimation accuracy, especially in the *z*-direction. The numerical results of the estimation errors are presented in Table 2. The root mean square of the errors (RMSE) is used here, including those of the errors in principal parameter directions, namely, the *x*, *y*, pitch, and yaw coordinates. In terms of RMSE, the error of our algorithm is lower than the other algorithms in all directions for a two-wheeled inverted pendulum robot.

### 5.2. Experimental Results and Analysis of Outdoor Structured Environment

In our outdoor experiments, we used algorithms such as lio_sam for comparison with our algorithm. Figure 15 shows our algorithm’s trajectory and environment map on Dataset Outdoor. Figure 16 compares estimated trajectories in the *x* and *y* directions on Dataset Outdoor, including our algorithm, fast_lio2, lio_sam, loam, odometry, and ground truth. Figure 17 and Figure 18 are the corresponding errors relative to the ground truth, respectively. Figure 19 shows the trajectory in the *z*-direction, which should be constrained around zero. As can be seen from the results, our algorithm continues to outperform the other algorithms in outdoor scenes.

Moreover, the accuracy improvement of our algorithm is more significant in the *z*-axis direction than in the *x*- and *y*-axis directions. There is no degradation scene in the outdoor experiment. The lidar feature points mainly come from the reflection of surrounding objects, which constrains the robot’s pose in the horizontal direction. In contrast, lidar provides relatively few feature points in the vertical direction due to the limitation of the vertical field of view. Our algorithm introduces ground constraints, which increase the pose constraints of the robot in the vertical direction and improve the localization accuracy of the system, especially in the *z*-direction. Meanwhile, we quantitatively analyzed the performance of our algorithm and other algorithms in the direction of the principal parameters on the outdoor dataset in terms of RMSE. The results are shown in Table 3, indicating that the error of our algorithm in the direction of principal parameters is lower than that of the other algorithms.

Our indoor and outdoor experiments show that the non-full constraint factor proposed by our algorithm naturally introduces the pre-integration constraint of the encoder while adding a ground constraint in the form of the random constraint model, which is more in line with the actual motion of the two-wheeled inverted pendulum robot, resulting in the improved positioning and mapping accuracy of the system.

## 6. Conclusions

This paper proposes a novel Lidar–IMU–Odometer coupled framework for localization and mapping for a two-wheeled inverted pendulum robot on approximately flat ground. We formulate a factor graph incorporating relative and absolute measurements from different sensors (including ground constraints) into the system as factors. In this coupled framework using a lidar–IMU–encoder system based on a factor graph, we introduce five types of factors and analyze the constraint dimension of each type of factor on the robot state node according to the motion characteristics of TWIP. For the non-fully constrained factors (the encoder pre-integration factor and ground constraint factor), we propose a new factor with constraints in order to add them naturally to the factor graph with the robot state node on SE(3). Experiments in indoor and outdoor environments validate the superior performance of our system compared to other state-of-the-art lidar-based SLAM algorithms.

This proposal paves the way for using constrained models to consider motion perturbations for specially-structured ground robot state estimation with different sensors. In the future, based on the work inof this paper we intend to realize the positioning and mapping of TWIP on undulating roads via the proposed LiDAR–IMU–Encoder coupling framework

## Figures and Tables

**Figure 1 sensors-22-04778-f001:**
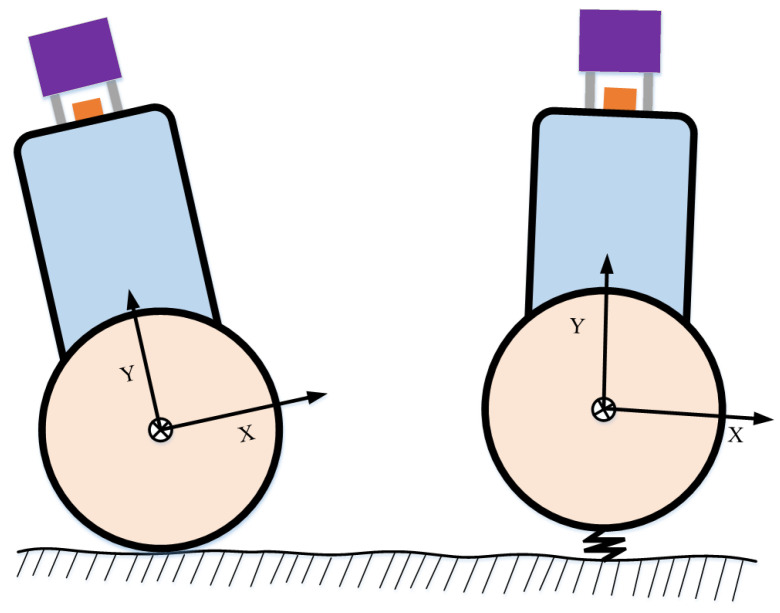
Constraint model: (**left**) deterministic constraints and (**right**) stochastic constraints, considering the motion perturbations addressed in this article.

**Figure 2 sensors-22-04778-f002:**
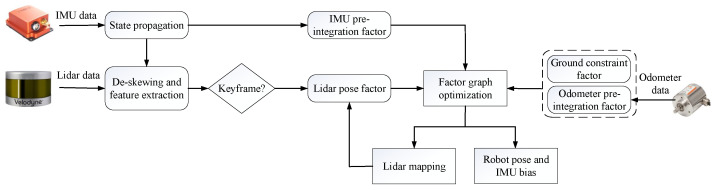
The flowchart of our algorithm.

**Figure 3 sensors-22-04778-f003:**
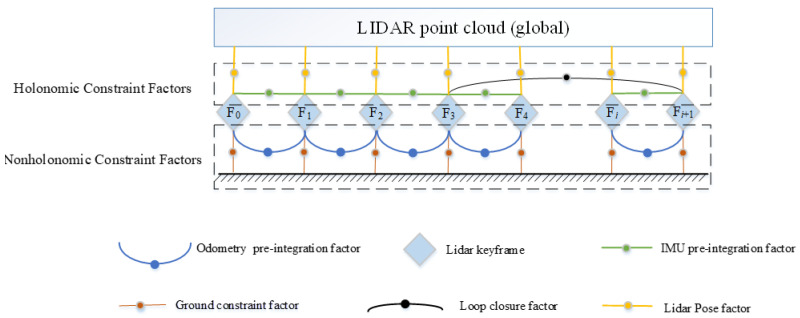
Factor graph optimization.

**Figure 4 sensors-22-04778-f004:**
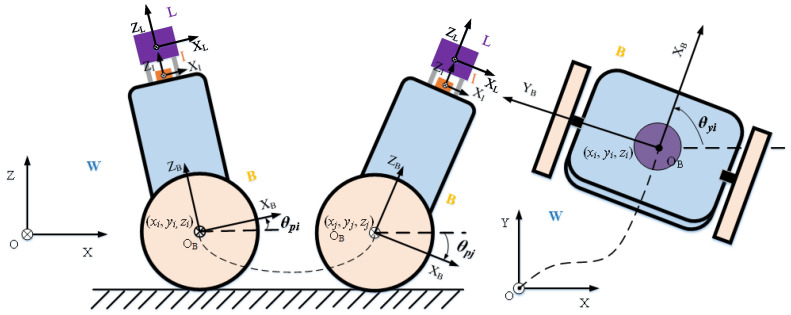
Simplified diagram of TWIP motion.

**Figure 5 sensors-22-04778-f005:**
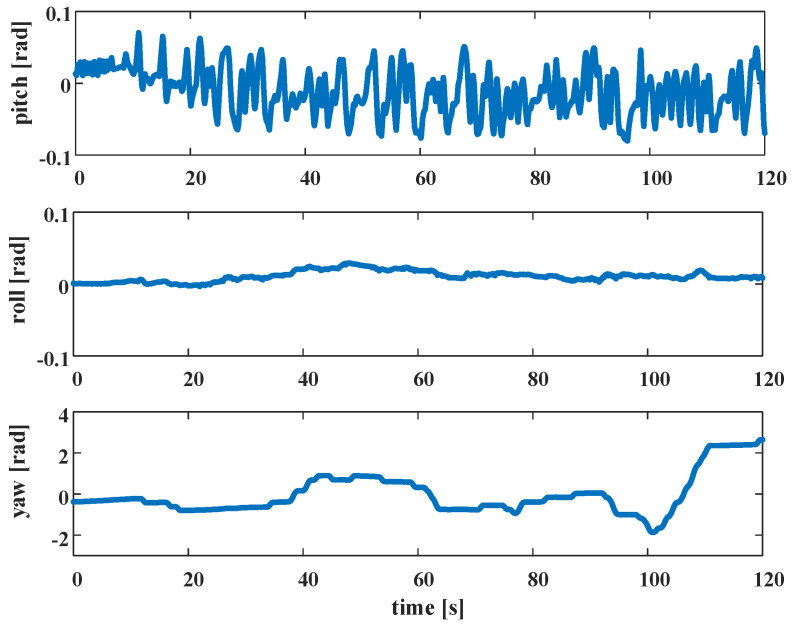
Pose of TWIP during movement (Euler angle).

**Figure 6 sensors-22-04778-f006:**
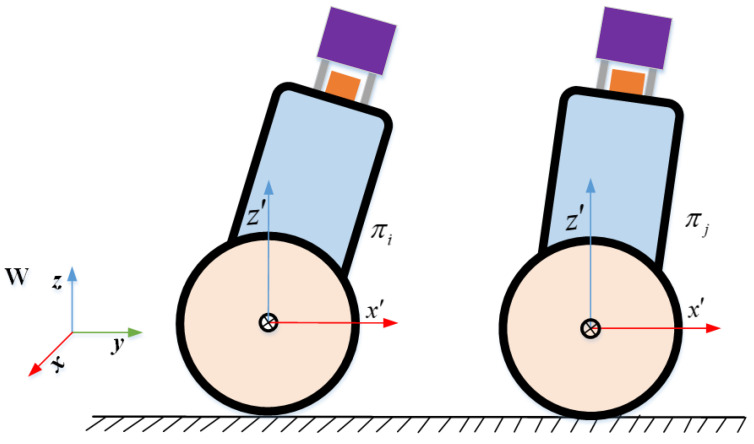
Motion of TWIP in Plane Coordinate System.

**Figure 7 sensors-22-04778-f007:**
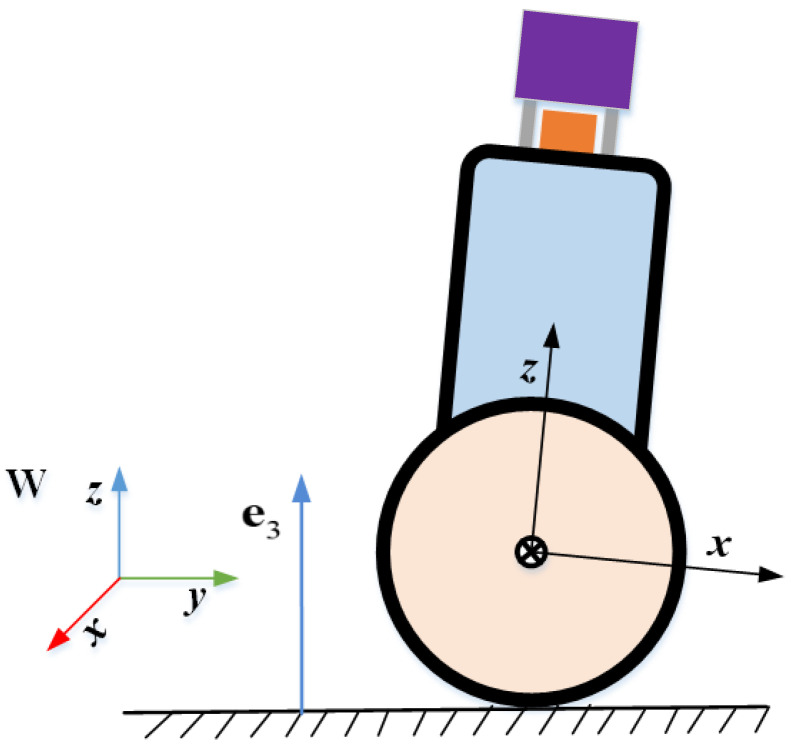
Ground constraints when TWIP is moving on approximately flat ground.

**Figure 8 sensors-22-04778-f008:**
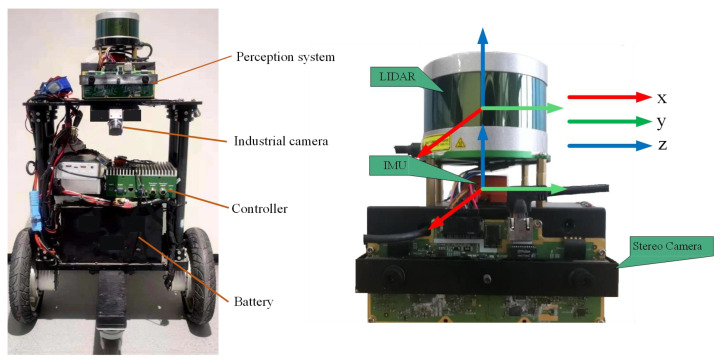
Experimental platform: Two-Wheeled Inverted Pendulum (TWIP) robot (**left**) and environment perception system (**right**).

**Figure 9 sensors-22-04778-f009:**
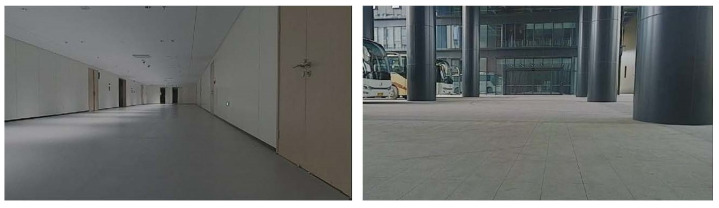
Experimental environment: indoor straight corridor (**left**) and outdoor structured environment (**right**).

**Figure 10 sensors-22-04778-f010:**
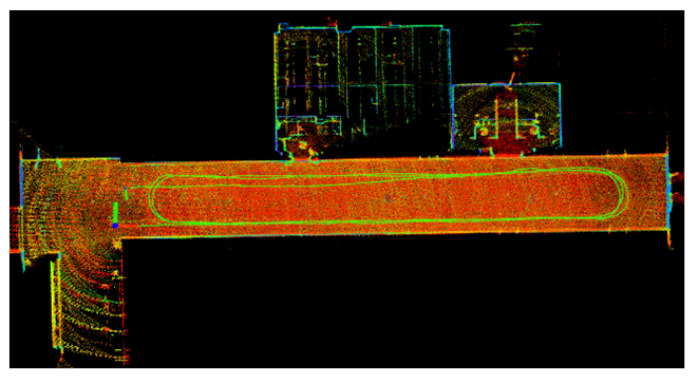
Localization and mapping results of our algorithm on Dataset Corridor.

**Figure 11 sensors-22-04778-f011:**
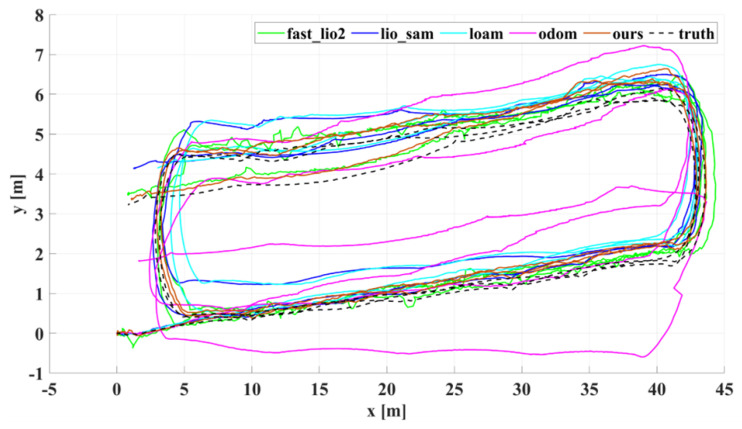
Dataset Corridor evaluation: projections of trajectories estimated by different algorithms on the xy plane.

**Figure 12 sensors-22-04778-f012:**
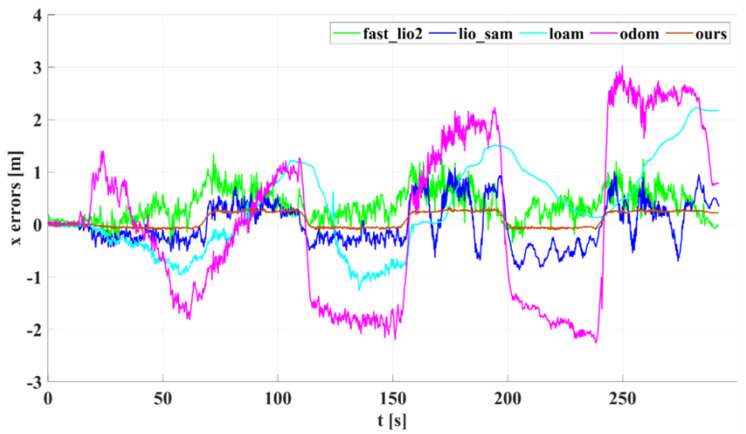
The *x*-axis estimation errors of different algorithms on Dataset Corridor.

**Figure 13 sensors-22-04778-f013:**
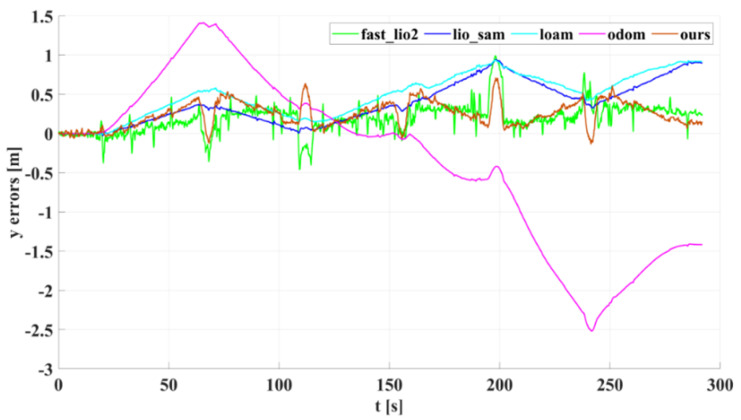
The *y*-axis estimation errors of different algorithms on Dataset Corridor.

**Figure 14 sensors-22-04778-f014:**
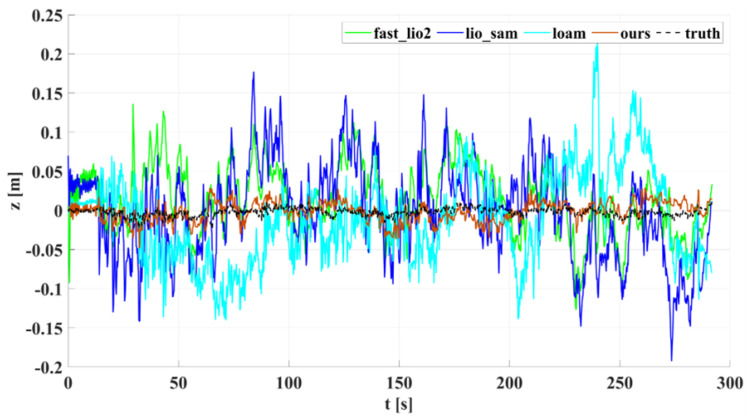
Dataset Corridor evaluation: projections of trajectories estimated by different algorithms on the *z*-axis.

**Figure 15 sensors-22-04778-f015:**
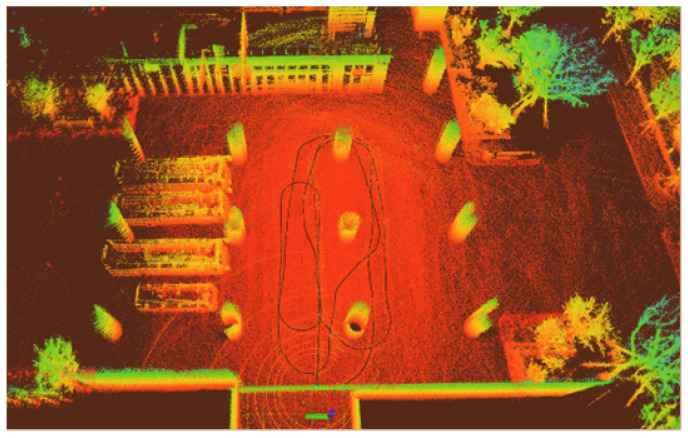
Localization and mapping results of our algorithm on Dataset Outdoor.

**Figure 16 sensors-22-04778-f016:**
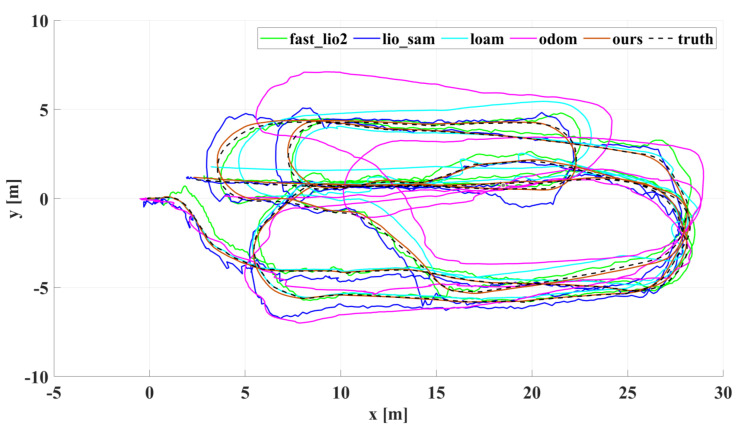
Dataset Outdoor evaluation: projections of trajectories estimated by different algorithms on the xy plane.

**Figure 17 sensors-22-04778-f017:**
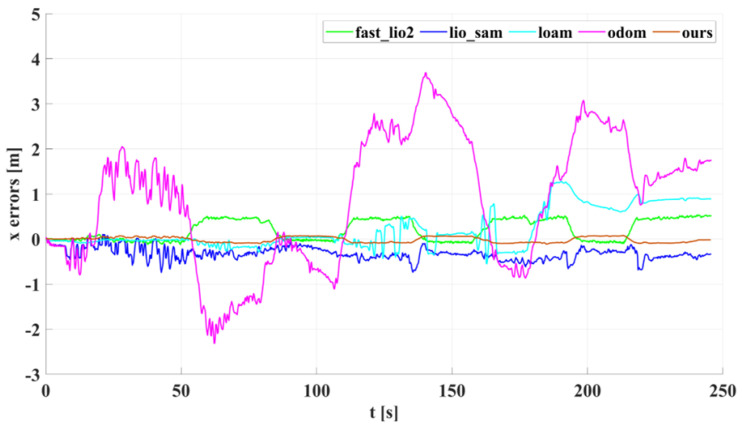
The *x*-axis estimation errors of different algorithms on Dataset Outdoor.

**Figure 18 sensors-22-04778-f018:**
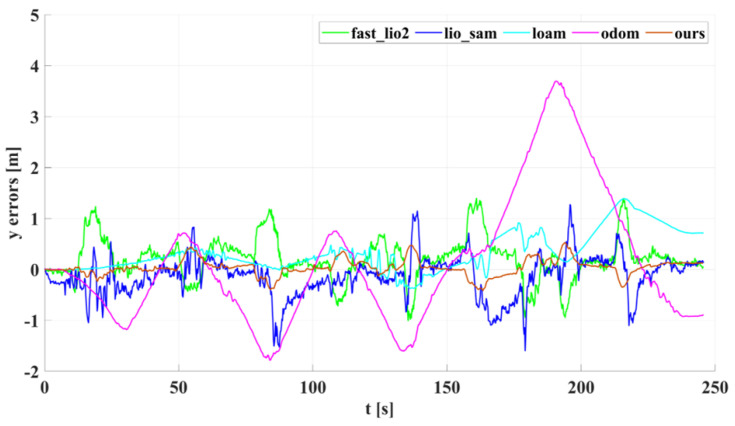
The *y*-axis estimation errors of different algorithms on Dataset Outdoor.

**Figure 19 sensors-22-04778-f019:**
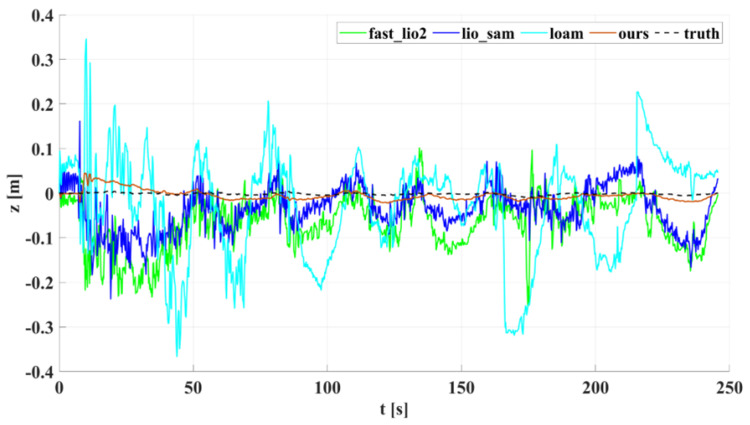
Dataset Outdoor evaluation: projections of trajectories estimated by different algorithms on the *z*-axis.

**Table 1 sensors-22-04778-t001:** State parameters and frame of TWIP.

Symbols	Descriptions
X,Z,O	The *x*-axis, *z*-axis, and origin of the world frame
$X_{B},\;Z_{B},\;O_{B}$	The *x*-axis, *z*-axis, and origin of the robot body frame
$X_{L},\;Z_{L},\;O_{L}$	The *x*-axis, *z*-axis, and origin of the lidar frame
$\left(x_{k},y_{k},z_{k}\right),k=i,j$	The position of the robot in the world frame at time *k* (in meters)
$\theta_{pk},\;k=i,j$	The pitch angle of the robot relative to the world frame at time *k* (in radians)
$\theta_{yk},\;k=i,j$	The yaw angle of the robot relative to the world frame at time *k* (in radians)

**Table 2 sensors-22-04778-t002:** The estimation errors (RMSE) of different methods on Dataset Corridor.

Dimension	Fast_lio2	Lio_sam	Loam	Odom	Ours
$x\left(m\right)$	0.4680	0.4111	0.9218	1.5468	0.1747
$y\left(m\right)$	0.2636	0.4695	0.5550	1.1035	0.2921
$pitch\left(rad\right)$	0.0287	0.0420	0.0590	−	0.0176
$yaw\left(rad\right)$	0.1740	0.1810	0.3170	0.6010	0.0970

**Table 3 sensors-22-04778-t003:** The estimation errors statistics (RMSE) of different methods on Dataset Outdoor.

Dimension	Fast_lio2	Lio_sam	Loam	Odom	Ours
$x\left(m\right)$	0.3113	0.3486	0.4770	1.6786	0.0635
$y\left(m\right)$	0.4549	0.4213	0.4999	1.2751	0.1625
$pitch\left(rad\right)$	0.0220	0.0281	0.0282	−	0.0172
$yaw\left(rad\right)$	0.2990	0.4900	0.6170	0.9368	0.1803

## Data Availability

Not applicable.

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
