# Peer review of "A Novel LiDAR–IMU–Odometer Coupling Framework for Two-Wheeled Inverted Pendulum (TWIP) Robot Localization and Mapping with Nonholonomic Constraint Factors"

_sensors, 2022, doi:10.3390/s22134778_

Round 1
Reviewer 1 Report
This paper studies a novel LiDAR-IMU-Odometer coupling framework for Two-Wheeled Inverted Pendulum(TWIP) robot localization and mapping with nonholonomic constraint factors. It has certain value for the application of TWIP robot in unknown environment. Here are some suggestions for articles:
- The text description in the author's article does not match the corresponding graphics on lines 284 and 290.
- The method proposed in the author's paper is relatively simple for the indoor test scene, and can be tested for the application scene of the two-wheeled inverted pendulum robot.
- Is the use of industrial cameras to detect markers proposed by the author in this paper for real-time pose estimation of robots? What is the effect on the overall algorithm?
- Authors can update some of the citations.
Reviewer 2 Report
1. The first paragraph of Introduction Part is too long so that author's point of view is not obvious.
2. Line 144: "ands" is unnecessary?
3. Line 208: "andhe"?
4. Line 294: The unit were missing
5. Line 290: Fig. 7?
Reviewer 3 Report
see file Review

Round 2
Reviewer 1 Report
This paper studies a novel LiDAR-IMU-Odometer coupling framework for Two-Wheeled Inverted Pendulum(TWIP) robot localization and mapping with nonholonomic constraint factors. It has certain value for the application of TWIP robot in unknown environment. However, there are some questions of the revised paper, just as follows:
(1) In Chapter 4.21, the author proposed that the feature point matching method used in this paper for SLAM conflicts with the closed-loop detection method using Euclidean distance proposed later. Plesse revise it.
(2) The node state described by the author in Figure 3 in Section 4.25 is not reflected in the figure.
(3) In the experiment, the author carries out offline optimization by using industrial camera to collect Aluko code and attitude sensor. However, this method is not mentioned as part of the algorithm framework in the paper. Please add related contents in the paper.
Reviewer 3 Report
There are two features in the paper
1. The traditional approach used in the integrated processing of measurements (data fusion) is based on filtering methods, for example, EKF. Here, instead of this, the factor graph optimization method is applied,
2. The authors use of Lie algebra in solving the orientation problem.
It is important to emphasize these features and discuss their advantages in the introduction, and make appropriate references to the literature in which these methods are discussed.
